# Persistent Cardiometabolic Health Gaps: Can Therapeutic Care Gaps Be Precisely Identified from Electronic Health Records

**DOI:** 10.3390/healthcare10010070

**Published:** 2021-12-31

**Authors:** Xiaowei Yan, Walter F. Stewart, Hannah Husby, Jake Delatorre-Reimer, Satish Mudiganti, Farah Refai, Andrew Hudnut, Kevin Knobel, Karen MacDonald, Frangiscos Sifakis, James B. Jones

**Affiliations:** 1Sutter Center for Health System Research, 2121 N. California Blvd, Suite 310, Walnut Creek, CA 94596, USA; husbyh@sutterhealth.org (H.H.); Mudigas@sutterhealth.org (S.M.); Jonesjb@sutterhealth.org (J.B.J.); 2Medcurio, Inc., Oakland, CA 94618, USA; wfs502000@yahoo.com; 3Formerly Sutter Health Research, 2121 N. California Blvd, Suite 310, Walnut Creek, CA 94596, USA; Jake.reimer@gmail.com (J.D.-R.); Farahrefai91@gmail.com (F.R.); 4Sutter Medical Group, Sacramento, CA 95833, USA; HudnutA@sutterhealth.org; 5Sutter Gould Medical Foundation, Modesto, CA 95355, USA; KnobelK@sutterhealth.org; 6Formerly AstraZeneca, Wilmington, DE 19897, USA; Kmac366@gmail.com (K.M.); FSifakis@gmail.com (F.S.)

**Keywords:** cardiometabolic conditions, health gaps, care gaps, medication adherence, electronic health record (EHR)

## Abstract

The objective of this study was to determine the strengths and limitations of using structured electronic health records (EHR) to identify and manage cardiometabolic (CM) health gaps. We used medication adherence measures derived from dispense data to attribute related therapeutic care gaps (i.e., no action to close health gaps) to patient- (i.e., failure to retrieve medication or low adherence) or clinician-related (i.e., failure to initiate/titrate medication) behavior. We illustrated how such data can be used to manage health and care gaps for blood pressure (BP), low-density lipoprotein cholesterol (LDL-C), and HbA1c for 240,582 Sutter Health primary care patients. Prevalence of health gaps was 44% for patients with hypertension, 33% with hyperlipidemia, and 57% with diabetes. Failure to retrieve medication was common; this patient-related care gap was highly associated with health gaps (odds ratios (OR): 1.23–1.76). Clinician-related therapeutic care gaps were common (16% for hypertension, and 40% and 27% for hyperlipidemia and diabetes, respectively), and strongly related to health gaps for hyperlipidemia (OR = 5.8; 95% CI: 5.6–6.0) and diabetes (OR = 5.7; 95% CI: 5.4–6.0). Additionally, a substantial minority of care gaps (9% to 21%) were uncertain, meaning we lacked evidence to attribute the gap to either patients or clinicians, hindering efforts to close the gaps.

## 1. Introduction

Cardiometabolic (CM) health is a dominant focus of secondary prevention management in primary care [1,2] The persistence of elevated CM clinical measures (e.g., elevated blood pressure, HbA1c, or low-density lipoprotein cholesterol (LDL-C)), denoted as health gaps, and related risk mediators substantially increase the risk of preventable morbidity and mortality [3,4,5,6,7]. Typically, a health gap is defined as a clinical parameter that is outside of the optimal range as defined by clinical practice guidelines (CPG). Separately, care gaps are defined as present if an action is not taken to close the health gap. Care gaps can span a range of issues. This study focused on therapeutic care gaps that are related to medication orders and adherence (referred to herein as “care gaps”). Physicians and patients can each contribute to the existence of therapeutic care gaps [8,9], e.g., when a clinician or care team member fails to prescribe CPG-recommended treatments (e.g., lab order or medication prescription) to close a health gap, or when a patient is prescribed but does not retrieve a medication intended to close a health gap. Relatively little is known about the extent to which health gaps are attributed to patient- or to provider-related care gaps.

Structured electronic health record (EHR) data are particularly well-suited to identify CM health gaps for control of blood pressure (BP), lipids, and hemoglobin A1c and can be used to facilitate individual- and population-level care. Automated algorithms can be used to extract clinical practice guideline-based outcomes, whether a patient has a health gap, and possible causes of the health gap(s) [10].

Surveillance without more specific actions to close CM health gaps does not necessarily translate to better health outcomes [11,12]. The ability to close health gaps is mediated, in part, by clinical-level (e.g., knowledge, communication, familiarity of guidelines) and patient-level factors (e.g., environment, health behaviors, genetics, etc.). However, the interactions that can occur among these factors can make it difficult to decide if an action is required for the patient, the physician, or both, which can result in delayed medication intensification or persistent clinical inertia and health gaps [13,14,15,16]. EHR structured data may often be sufficient to know if an intervention is needed but may also be lacking sufficient information to reveal what specific actions could be taken to improve patient-level outcomes. As medication dispense data are now more readily available in EHRs, it is possible to track if a patient retrieved their medication and to derive a medication adherence measure. However, little is known about the extent to which medication adherence data and use of other structured data can help to improve patient outcomes. 

The objective of this study was to determine the extent to which structured EHR data can be used to readily identify health and therapeutic care gaps for CM patients. In particular, we selected medication adherence as a disease modifier to help understand if structured EHR data can provide insight into whether the underlying care gap is attributable to the physician or to the patient, as this attribution provides a basis for acting to close the gap. 

## 2. Materials and Methods

### 2.1. Study Design and Population

In this retrospective analysis, a 2-year look-back period (i.e., 1 October 2013–30 September 2015) was used to define eligibility as a primary care patient. A subsequent 2-year period (1 October 2015–30 September 2017), denoted as the “Health Gap and Therapeutic Care Gap Assessment Period”, was used to identify health gaps and medication adherence status. Care gaps were assessed using structured data (e.g., diagnosis, medication order, laboratory results, etc.) in the 6-month period following identification of a health gap (Figure 1). 

The study was conducted at Sutter Health, a large not-for-profit health system in Northern California comprising 24 hospitals, 272 outpatient primary care practices, and approximately 1656 primary care physicians (PCPs), excluding pediatricians. We confined analyses to patients who had a Sutter primary care provider (PCP) and met the following eligibility criteria: (1) During a two-year look-back period (i.e., 1 October 2013–30 September 2015) they (i) had at least two clinical encounters with a Sutter PCP and (ii) met diagnostic criteria for hypertension, hyperlipidemia, or diabetes (details in Appendix A File 1). (2) During the health gap and care gap assessment period (i.e., 1 October 2015–30 September 2017) patients had (i) at least one CM-related clinical measure and one clinical encounter with a Sutter provider following the identification of a health gap or, for those without a health gap, following the date of the first biometric measure result in the EHR (i.e., the index date for those who had no health gap) and (ii) did not die in the 6-month post-period following the identification of a health gap. Figure 2 illustrated how the patients met eligibility criteria.

### 2.2. Data Sources

Only structured EHR data fields were used in this study and included the following: (1) encounter date and type; (2) ICD codes for encounter diagnoses, laboratory, or procedure orders; (3) clinical and laboratory measures relevant to CM disease risk factors, including body mass index (BMI), BP, lipids (i.e., low-density lipoprotein cholesterol (LDL-C), triglycerides, etc.), fasting blood glucose, HbA1c, and estimated glomerular filtration rate (eGFR) that may be modifiable or have a bearing on risk management; (4) patient sex, age, and race; and (5) medication dispense data provided by Surescripts. Dispense data were from 24,000 participating pharmacies providing coverage on more that 80% of medications dispensed for the Sutter Health primary care population. On a nightly basis, historic medication dispense data were retrieved from participating pharmacies for all patients with a scheduled encounter with a Sutter Health physician on the following day. 

### 2.3. Study Outcomes: Health and Care Gaps

The primary outcomes for the study were health and care gaps. In order to identify the health gaps for hypertension, dyslipidemia, and diabetes, we began by translating corresponding clinical guidelines [17] into operational criteria that can be applied to structured EHR data. 

The operational criteria were developed for health gaps as follows: a health gap exists when a clinical measure is at or above the specific threshold (e.g., systolic blood pressure [SBP] ≥ 140 mmHg) as defined by CPG criteria, which were active during the study period (see Table 1) [18]. Two sequential elevated BP measures obtained during encounters on separate days were used to define a BP health gap, consistent with American College of Cardiology (ACC)/American Heart Association (AHA) guidelines. One elevated measure was used to define an HbA1c (A1c ≥ 7) or LDL-C health gap (LDL ≥ 190, coronary heart disease (CHD) risk > 20%) [19,20]. While there may be nuances in these thresholds dependent on various personalized patient factors, for the purposes of algorithm development, the cutoffs for gap identification were simplified and standardized based on the active clinical guidelines during the study period. 

The first encounter where a qualifying health gap was identified in the “Health Gap and Therapeutic Care Gap Assessment Period” was denoted as the index date (Figure 1). If there was no health gap, the index date was defined as the first encounter when the qualified clinical measure(s) was obtained, or the first PCP encounter for those who did not have a qualified measure. As two instances of elevated BP were required to define a health gap, and the index date was defined by the date of the second elevated BP measure to simulate how the gap identification would be managed in clinical practice. 

Among patients who had a health gap, we created operational criteria to identify the corresponding care gaps. Medication adherence is usually not specifically articulated in the guidelines; however, it is an important factor to drive clinical actions (e.g., education, prescription, monitoring). We incorporated medication adherence into clinical treatment guidelines to create the operational criteria for identification of care gaps (Table 1) validated by Sutter Health clinicians.

Medication adherence, measured using dispense data, was extracted from the Sutter EHR for the 12-month period preceding each office encounter to calculate the proportion of days covered (PDC) for a specific medication [21]. PDC was calculated as days covered by the filled prescription period divided by 365 days (i.e., 1 year) if the medication was prescribed for more than 1 year, or divided by the number of days between first prescription and index date. If there were no medication dispense data for a given medication order, the patient was defined as not retrieving the medication. If there were medication dispense data, then the medication adherence status of a given patient was categorically defined as PDC values of ≥80% (adherent) and <80% (non-adherent) [22]. 

A care gap was identified within the 6-month post-period following identification of a health gap, and was assessed both with and without adherence information, because not all healthcare systems have medication adherence available in the EHR. In the absence of medication adherence data, a care gap was defined as the absence of any EHR record of a medication order or a medication adjustment (e.g., increased dose, change or add medications) for any patient with a health gap. When medication adherence data were present, a care gap was defined as follows: (i) for adherent patients with a health gap, a care gap existed if there were no record of a medication adjustment (i.e., added or intensified medication) being made to close the health gap; (ii) for non-adherent patients with a health gap, a care gap existed if there were no record that a previously prescribed medication was re-ordered or that no new medication was ordered; and (iii) for patients with a health gap who did not retrieve their medication (no dispense data present in the EHR), a care gap existed if there was no evidence of a new order to restart the medication. 

### 2.4. Statistical Analysis

Using structured EHR data, we estimated the following: (1) the prevalence of health gaps, (2) medication adherence for patients with a health gap, and (3) the proportion of patients with care gaps. 

Summary statistics were calculated for patients who met eligibility criteria for the study, and for each CM condition that included demographics, BMI, and comorbidities (measured by Charlson Comorbidity Index, or CCI) [23], as well as the prevalence of a health gap for each biometric measure (Table 2). 

Medication adherence level was categorized into three groups: (1) if medication was ordered and the patient retrieved the medication, then PDC was calculated as noted above, and we further stratified the patient into adherent (PDC ≥ 80%) or non-adherent (0 < PDC < 80%); (2) if medication was ordered and the patient did not retrieve the medication, then the patient was defined as non-adherent, but was denoted as medication not retrieved; and (3) if medication was not ordered by the clinician, the patient status was defined accordingly. 

Further analyses were limited to patients with complete data (last row in Table 2, patients who had qualified lab measures and dispense data). Health gap status (yes/no) was the outcome. We assessed the extent to which health gaps are explained by a four-category adherence variable using logistic regression, reported by the odds ratio (OR) and 95% confidence interval (95% CI) (Table 3) among the population for whom dispense data were available. The regression modeling was completed, adjusting for the age at index date, sex, race (20 options categorized as African American, Asian, White, and other), ethnicity for Hispanic status yes, no, unknown), BMI (<25, 25–29.99, 30–34.99, 35+, unknown), smoking status (never, quit, smoking/passive, unknown), alcohol intake status (yes, no, unknown), CM disease severity defined by the number of CM conditions a patient had, and CCI. 

We further tested whether the association between medication adherence and health gaps varied among race groups by adding an interaction term between race group and medication adherence. Subgroup analyses were conducted to estimate OR and 95% CI for medication adherence and each health gap for each race group. Results are summarized in Table A1.

Care gap analysis was limited to patients who had a health gap. The criteria described in Table 1 were used to identify therapeutic care gaps for each health gap. A classification of “uncertain” was assigned to a care gap when there was a lack of evidence in the structured EHR data to document a physician’s efforts to close a health gap. Descriptive statistics were calculated to estimate the prevalence of care gaps, stratified by the medication adherence level (Table 4). Summary statistics by patient care gap status (i.e., no care gap, care gap, uncertain) on demographics, CCI, smoking status, baseline biometric value, and medication adherence are summarized in Table A2, Table A3 and Table A4.

All analyses were performed using SAS Enterprise Guide 7.1 (Cary, NC, USA). The Sutter Health Institutional Review Board approved this study.

## 3. Results

A total of 252,181 eligible Sutter Health adult primary care patients aged 35+ years were identified (Table 2) with one or more diagnosed CM diseases (i.e., hypertension, hyperlipidemia, and diabetes).

Among the eligible primary care patients, 76% had hypertension, 75% had hyperlipidemia, and 23% had diabetes, and 17.8% had all three CM conditions. The prevalence of each CM condition varied by demographic features and health behaviors (Table 2).

Among all eligible patients, 95.4% (*n* = 240,582) had one or more CM clinical measure during the health gap follow-up period. Of these, 54.3% had at least one health gap. The prevalence of health gaps was 43.8% for hypertension, 57.2% for diabetes, and 33.5% for hyperlipidemia (Table 2). Among patients with qualified corresponding biometric measures, 81% had medication dispense data, varying from 75% to 85% by CM condition (Table 2).

Health gaps are highly associated with medication adherence status. Among patients who had a health gap, 26–37% were classified as adherent (i.e., ≥80% PDC) (Table 3). Compared to adherent patients, patients with a PDC < 80% (OR: 3.23, 95% CI: 2.50–4.39) were more likely to have a hyperlipidemia health gap, which was even more pronounced for those with no evidence of a medication order (OR: 5.80, 95% CI: 5.58–6.02) (Table 3). Similarly, the elevated risk of HbA1c health gap was observed for patients with PDC < 80% (OR: 1.15, 95% CI: 1.08–1.33) and more than five-fold higher risk for those with no evidence of a medication order (OR: 5.66, 95% CI: 5.35–5.99) (Table 3). The odds of having a BP health gap were slightly elevated for patients with a PDC < 80% but not statistically significant (OR: 0.97, 95% CI: 0.91–1.02). More generally, those who did not retrieve their medication and those with no medication order had elevated adjusted odds for a health gap for all three diagnosed diseases (Table 3).

The interaction term between medication adherence and race group was statistically significant for all three health gaps (*p*-values < 0.001). Subgroup analyses revealed that the level of the association (i.e., OR) between medication adherence and health gap varied among different race/ethnicity group, but the direction of the association (OR > 1 or OR < 1) remained the same across all racial groups (see Table A1).

Among patients with a health gap, 54.1%, 33.3%, and 64.6% also had a therapeutic care gap associated with managing BP, HbA1c, and LDL-C, respectively (Table 4). Within each medication adherence classification, the care gap status can be further grouped into yes, no, and uncertain (Table 4). For patients with a BP health gap, 31.5% patients were adherent to medication. Of those who were adherent to BP medications, therapeutic care gap status can be determined for 60% of patients (37.9% with therapeutic care gap and 22.1% without), and 40% where the care gap status was uncertain. Among patients who had an LDL health gap and were adherent to medication, 50.8% had a therapeutic care gap, 18.5% had no therapeutic care gap, and 30.8% were uncertain, which was similar for patients with an HbA1c health gap who were adherent to treatment (i.e., 47.8% with therapeutic care gap, 23.9% no therapeutic care gap, and 28.3% uncertain). However, therapeutic care gap status was uncertain for a significant portion of patients and especially among those with PDC < 80% who had hyperlipidemia (55.9%) and diabetes (70%) (Table 4). Among patients who did not retrieve a medication, 58.7%, 39.1%, and 15.6% had a therapeutic care gap identified for BP, LDL, and HbA1c management, respectively. Among patients with a health gap and no medication order, a care gap existed among 49.9% for HbA1c management and 82.1% for LDL management. Among those with no medication order, that is, no physician effort was observed to close a BP health gap, 100% of patients in this subgroup had a therapeutic care gap (Table 4) and the therapeutic care gap was attributed to a physician.

Among patients with hypertension, care gaps were more common for those with more CM conditions, those who were older (75+ years older), Black, had higher BMI, more comorbidities and did not retrieve their medications (Table A2). Among patients with dyslipidemia, care gaps were more common for those who were male, 55–64 years of age, Black, Hispanic, had a higher BMI, were smokers and had higher baseline LDL-C (Table A3). Among DM patients, care gaps were more common for males, <65 years old, and patients who did not retrieve medication (see Table A4).

## 4. Discussion

In this retrospective data analysis, the study team was able to identify all cardiometabolic health gaps and the majority of therapeutic care gaps using structured EHR data. Cardiometabolic therapeutic care gaps were common and varied significantly by patient medication adherence levels. The prevalence of therapeutic care gaps is distinctly different for patients who appear to not have retrieved medication versus patients who are adherent to their medication. Moreover, uncertainty in determining therapeutic care gap status varied by patient adherence level. For a significant minority of patients, the study results reveal that the use of structured EHR data alone is not sufficient to determine whether the patient, clinician, or both are responsible for a therapeutic care gap. In such instances, the reasons that a clinician did not initiate/titrate or modify medications may be documented in unstructured fields in the EHR. This suggests that progress notes and/or patient self-reported data may be useful in supporting physicians to efficiently and effectively manage care gaps, although exploring this possible solution was out-of-scope for this paper.

In the 6 months following identification of a health gap, this study found no evidence that medications were managed for more than half of patients with elevated blood pressure and LDL-C, and a third with an elevated HbA1c. This may be explained by a lack of recognition of a health gap by the physician, or by the physician’s perception of the efficacy of the medication, or by a lack of patient engagement in using the medication. The insufficient management by a physician is known as a physician care gap in this study, and the lack of patient use of the prescribed medication, known as a patient care gap, can be inferred by using EHR data to estimate patient treatment adherence. Though it is not possible to completely separate them, the actions to close a care gap are likely distinctly different when a patient appears to be non-adherent versus adherent; therefore, measuring adherence may shed light on challenges and strategies for managing therapeutic care gaps. 

In the case of a medication not being ordered, the onus is most clearly on the physician and, therefore, the actions to close the care gap may be most straightforward and involve the least physician effort. For example, automated algorithms can be used to auto-pend orders for physician review when medications have not been ordered for a defined health gap. Non-adherence (i.e., medication not retrieved) can also be identified from structured EHR data and, once detected, the actions physicians can take are reordering the existing medications and/or educating patients on the importance of medications. 

For patients with low adherence (PDC < 80%) the decision on appropriate actions to close a health gap is more nuanced. This study reveals that substantial uncertainty (55.9% to 70%) in care gaps status exists for low adherent patients with LDL and HbA1c health gaps. Further analysis shows that the uncertainty occurred mostly in patients with multiple therapeutic medications, where efforts to intensify medications are not straightforward to identify, in part because of definitional challenges, such as whether a care gap-closing action has been taken when at least one medication is changed to a different one but other medications remain the same. The lack of this level of detail in guidelines leads to physician’s individual judgement, which hinders standardization across different physicians and practices. Regardless of the attribution of the care gap, when low adherence is present, some automated or semi-automated actions may be possible, such as proactive outreach via tailored communications from the physician or an invitation to meet with a pharmacist, where such a resource exists.

Paradoxically, this study found that more physician work is required for adherent patients. For example, about 75% of diabetes patients with a persistent care gap were adherent to their prescribed medication, where it appears that the medication needs of the patient were not adequately managed. That is, there is no evidence in the patient’s EHR that the physician made an effort to determine if medication switching, titrating, or additions would mitigate the health gap. 

In addition, we found that the prevalence of therapeutic care gaps varied by race/ethnicity groups. African American patients had a slightly higher prevalence of care gaps for hypertension and dyslipidemia, a finding consistent with prior studies on the initiation/intensification of medication [24,25,26]. Using EHR data to identify all patients with health gaps and creating targeted strategies to manage patients with different levels of medication adherence might reduce the disparity. 

Physician lack of recognition, lack of follow-up, or simple time constraints [27] and administrative workload [28,29] may explain why a care gap persists. Medication management requires recognition of the health gap and then, in the moment, the need to review the patient record to understand what has been tried, a somewhat precise recall of relevant guideline knowledge, and a decision about the options that are logically worth discussing and testing. Structured and unstructured information may have to be reviewed from previous encounters. Then, the patient’s needs may have to be assessed to get to an optimal decision. The effort required to close the care gap may be more demanding than time will allow, especially if there are challenges related to patient engagement and self-management. Incomplete assessment of patient needs may simply be explained by how little time a physician has with a patient to understand needs and preferences [30,31,32,33,34,35]. As an alternative, pharmacists and other members of the care team could play a role in following up with patients to improve closure of care gaps [36,37].

Patient care gaps may persist because of patient-level factors that are directly under the patient’s control (e.g., low treatment adherence), or completely out of the patient’s control (e.g., lack of insurance coverage) [38,39]. In reality, the lack of a structured and efficient approach to document patient’s needs and indicators of medication adherence may result in a default attribution of medication care gaps to the clinician [26,40,41]. These problems in closing care gaps are not new. The need to support physicians to make rapid and reasonable decisions on what to do next and, in particular, what to discuss with the patient, can be met by assembling and displaying relevant clinical data and patient-reported data.

In summary, to improve EHR-facilitated patient health management, three essential components of information are necessary: efficient presentation of EHR structured data (e.g., potentially through visualization), information extracted from patient-reported data (e.g., physical activity, mental fitness, self-efficacy, insurance, etc.) or progress notes (i.e., patient–clinician communication regarding the treatment plan), and translation of clinical guidelines (i.e., with relevant contextual data and risk calculators to assess guideline applicability) to inform personalized recommendations. Clinical decision support tools that contain these three components might be a solution to improve quality of care by closing both health and care gaps [42,43,44,45]. 

There are several limitations to this study. First, the data used for the analysis came from a single health care system with a demographically diverse population. Second, the study did not consider other care gaps (e.g., screening, referral to dietician, diabetes educator, lifestyle consultant, etc.) for which the same challenges in actionable documentation exist. However, the described approach to using EHR data can be applied to other types of care gaps. Third, EHR data may be incomplete to fully decide if health gaps were resolved or care gaps were closed as the EHR may not capture whether patients received care or medication elsewhere. The missing data problem may be addressed, in part, as Epic’s Care Everywhere and similar EHR functions increase access to patient encounter data independent of the provider group. Fourth, this study did not use free-text data (i.e., progress notes) as the team were not resourced to extract and validate treatment plans, patient behavior, and the clinician’s justification using natural language processing (NLP). Nonetheless, unstructured data hold the promise of revealing specific actionable causes of care gaps. 

## 5. Conclusions

Structured EHR data can identify health gaps and a majority of care gaps. Actions to close health gaps vary significantly at different medication adherence levels. Using structured EHR data to identify health gaps and measure medication adherence can be a first step in identifying factors attributable to care gaps and thereby help guide the direction for clinical action. Uncertainty about the clinical actions best suited to close health gaps can be minimized by three essential components: efficient presentation of EHR structured data, integration of patient-reported data or progress notes to indicate patient preferences, and translation of clinical guidelines to inform personalized recommendations. 

## Figures and Tables

**Figure 1 healthcare-10-00070-f001:**
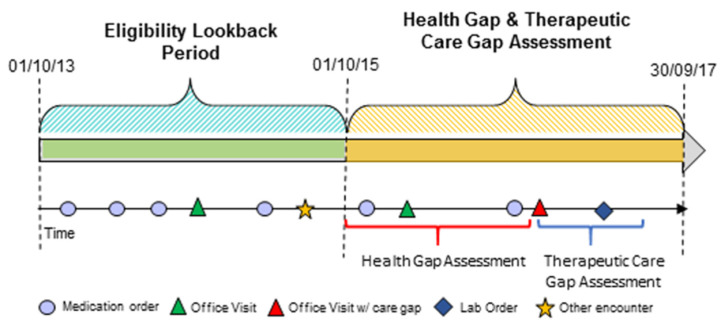
Retrospective use of EHR data to identify eligible primary care patients, as well as health and care gaps.

**Figure 2 healthcare-10-00070-f002:**
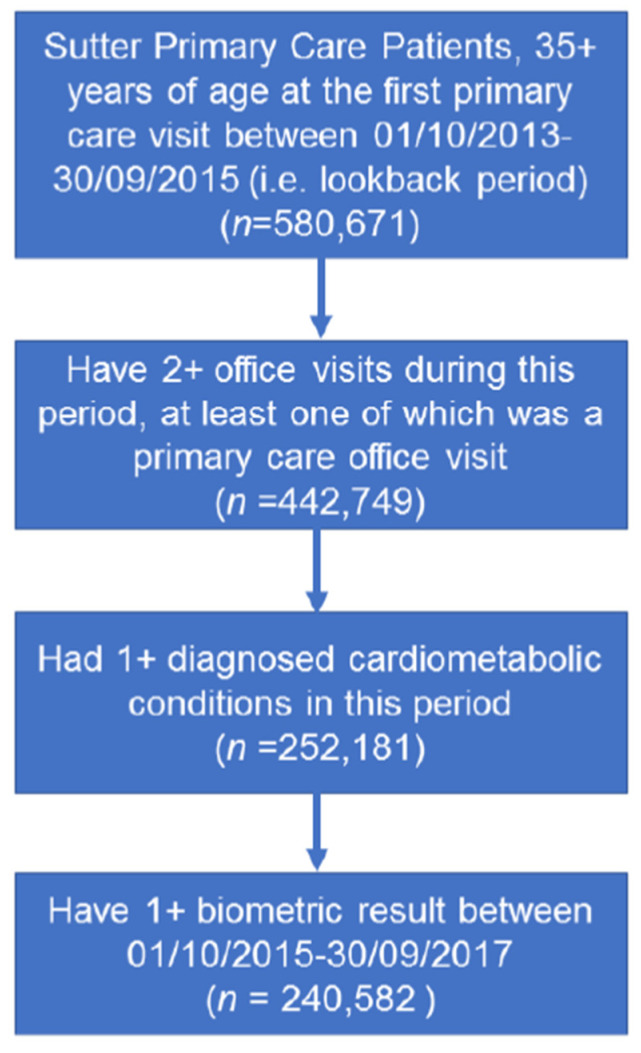
Flowchart for study cohort identification. Flowchart for inclusion of Sutter primary care patients 35+ years of age to identify individuals with a cardiometabolic diagnosis and related health gap.

**Table 1 healthcare-10-00070-t001:** Criteria for defining health gaps and therapeutic inertia care gaps.

	Disease	Health Gap	Criteria ^a^
Health Gap Criteria	Hypertension	Yes	Diastolic BP ≥ 90 mmHg OR systolic BP ≥ 140 mmHg in two consecutive clinical encounters
	Dyslipidemia	Yes	If CHD 10-year risk ^b^ > 20% then LDL health gap was defined as ≥ 100 mg/dL If there were 2+ risk factors ^b^ or the 10-year risk ≤ 20% then the LDL health gap was defined as ≥ 130 mg/dL If there were 0–1 risk factors then the LDL health gap was defined as ≥ 160 mg/dL
	Type II Diabetes	Yes	HgA1c ≥ 7.0%.
Therapeutic Care Gap Criteria	Medication Order Status Prior to Gap Identification	Care Gap Present	Criteria of Actions Taken to Close Therapeutic Inertia
	Medication ordered by physician and retrieved by patient with PDC ≥ 80%	No	Treatment is intensified by increasing the dose of at least one medication or by adding a second medication to the existing regimen
		Yes	Medication is the same as the pre-health gap medication(s), or no medication was prescribed in the post health gap period
		Uncertain	Part of the medication regimen has been changed or, for medications that are not changed, doses are the same
	Medication ordered by physician and retrieved by patient with PDC < 80%	No ^c^	Reorder the existing medication ^d^
		Yes	There was no continuation of the medication order
		Uncertain	Patient continued to have low adherence and it is unknown whether the physician had a discussion with the patient to improve adherence
	Medication ordered by physician but not retrieved by the patient	No	A medication was re-ordered in the post health gaps period ^d^
		Yes	No medication was prescribed in the post health gap period
	No Medication ordered	No	A medication was ordered in the post health gap period
		Yes	No medication was ordered in the post health gap period, and it is uncertain if the patient refused to take a medication or discontinued as a primary care patient with the physician

^a^—Criteria are consistent with the American College of Cardiology (ACC)/American Heart Association (AHA) guidelines. ^b^—Factors used to estimate CHD risk: age, total cholesterol, smoking status, HDL, systolic BP, and antihypertensive treatment. The formula can be found in ATP III guidelines. ^c^—Assumes that the physician discussed adherence with the patient, but the patient may have also acted on their own to improve treatment adherence. ^d^—Assumes that the physician recognized the patient self-care gap and acted to close the care gap with a new prescription order with or without contacting the patient.

**Table 2 healthcare-10-00070-t002:** Demographic and clinical characteristics of primary care patients 35+ years old diagnosed with hypertension, hyperlipidemia, or diabetes.

Baseline Status	Category	Percent of Total Population (*n* = 252,181)	Percent of the Total Source Population with a Diagnosis		
			Hypertension	Hyperlipidemia	Diabetes
			*n* = 191,876	*n* = 188,366	*n* = 57,534
Total		100%	76%	75%	3%
Diagnosed CM Diseases	1	43.6%	28.7%	28.1%	2.2%
	2	38.6%	47.9%	48.1%	19.9%
	3	17.8%	23.4%	23.8%	77.9%
Gender	Female	54.6%	55.0%	52.0%	51.0%
	Male	45.4%	45.0%	48.0%	49.0%
Age	35–44	7.6%	6.3%	6.1%	5.0%
	45–54	20.8%	18.4%	18.6%	17.0%
	55–64	26.6%	25.7%	26.6%	25.6%
	65–74	25.0%	26.4%	27.1%	28.2%
	75+	19.9%	23.2%	21.7%	24.2%
Race	White	64.6%	66.3%	64.4%	55.2%
	Black	3.5%	4.0%	3.2%	5.4%
	Asian	13.5%	11.7%	14.1%	17.0%
	Other	18.4%	18.1%	18.3%	22.4%
Hispanic	Yes	9.6%	9.9%	9.7%	14.3%
	No	90.4%	90.1%	90.3%	85.7%
BMI	<25	26.1%	24.0%	24.8%	16.1%
	25–29	36.9%	35.8%	37.9%	31.1%
	30–34	21.8%	23.1%	22.5%	26.9%
	35+	14.7%	16.6%	14.3%	25.5%
	Missing	0.6%	0.6%	0.5%	0.5%
Charlson Score	0	68.6%	65.1%	67.0%	47.9%
	1–2	26.7%	29.5%	28.0%	42.3%
	3+	4.6%	5.4%	5.0%	9.8%
% with qualified biometric measure ^a^	Yes	95.4%	98.2%	91.9%	95.1%
Health Gap ^b^	Yes	54.3% ^c^	43.8%	33.5%	57.2%
% of patient with dispense data	Yes	81%	85%	75%	78%

^a^. For hypertension, two consecutive BP measures were required, for others, only one measure was required. ^b^. Denominator is those in a, that is, those who had qualified corresponding biometric measures. ^c^. % with at least one health gap.

**Table 3 healthcare-10-00070-t003:** Logistic regression model-based estimates of the odds ratio for the relationship between medication adherence, as defined by proportion of days covered (PDC) and patient health gap status with and without adjustment for other covariates ^a^.

Patient Medication Adherence ^b^	Hypertension Health Gap (*n* = 83,033)			Hyperlipidemia Health Gap (*n* = 54,647)			Diabetes Health Gap (*n* = 31,297)		
	PCT	Unadjusted	Adjusted ^a^	PCT	Unadjusted	Adjusted ^a^	PCT	Unadjusted	Adjusted ^a^
80%+ PDC ^b^	36.7% ^d^	Ref	Ref	25.7% ^d^	Ref	Ref	34.0% ^d^	Ref	Ref
0–79% PDC	8.2%	1.07 (1.02–1.12)	0.97 (0.91–1.02)	7.4%	3.21 (2.76–4.34)	3.23 (2.50–4.39)	11.6%	1.17 (1.06–1.30)	1.15 (1.08–1.33)
Patient did not retrieve medication	38.9%	1.44 (1.41–1.47)	1.23 (1.21–1.26)	26.6%	1.67 (1.61–1.74)	1.76 (1.69–1.83)	27.5%	1.25 (1.19–1.32)	1.30 (1.23–1.37)
No medication order ^c^	16.1%	1.48 (1.44–1.53)	1.25 (1.22–1.29)	40.4%	4.19 (4.05–4.33)	5.80 (5.58–6.02)	26.8%	5.62 (5.32–5.94)	5.66 (5.35–5.99)

^a^—adjusted for age, sex, race, BMI, number of CM conditions, ethnicity, smoking status, and alcohol status. ^b^—includes patients with hypertension (1.5%), diabetes (1.6%), and hyperlipidemia (1.3%) who did not have a Sutter medication order but did have a medication dispense claim. ^c^—includes no medication order or no medication dispense as evidence of a prescription order. ^d^—43.7% patients with BP health gap and with medication order had 80%+ PDC; 43.1% patients with LDL health gap and medication order had 80%+ PDC; and 46.4% patients with HbA1c health gap and medication order.

**Table 4 healthcare-10-00070-t004:** Prospective therapeutic care gap (no actions taken to close health gap) status among primary care patients 35+ years old diagnosed with hypertension, hyperlipidemia, or diabetes who have a disease specific health gap by medication adherence status.

Medication Adherence	Therapeutic Care Gap Present	HTN with BP Health Gap (*n* = 82,980) ^a^		Hyperlipidemia with LDL Health Gap (*n* = 41,405) ^b^		Diabetes with HbA1c Health Gap (*n* = 23,131) ^c^	
Overall	No	30.2%		26.5%		45.4%	
	Yes	54.1%		64.6%		33.3%	
	Uncertain	15.7%		8.9%		21.3%	
80%+ PDC ^d^	No	*n* = 26,172 (31.5%)	22.1%	*n* = 5617 (13.6%)	18.5%	*n* = 9579 (41.4%)	23.9%
	Yes		37.9%		50.8%		47.8%
	Uncertain		40.0%		30.8%		28.3%
0–79% PDC ^e^	No	*n* = 15,299 (18.4%)	46.7%	*n* = 3427 (8.3%)	12.1%	*n* = 3171 (13.7%)	17.0%
	Yes		37.5%		32.0%		13.0%
	Uncertain		15.8%		55.9%		70.0%
Patient did not retrieve medication	No	*n* = 29,524 (35.6%)	41.3%	*n* = 8697 (21.0%)	60.9%	*n* = 7212 (31.2%)	84.4%
	Yes		58.7%		39.1%		15.6%
No medication order	No	*n* = 11,985 (14.4%)	0%	*n* = 23,664 (57.1%)	17.9%	*n* = 3169 (13.7%)	50.1%
	Yes		100%		82.1%		49.9%

^a^—Out of 83,033 HTN patients who had BP health gap, 99% had Surescripts dispense data and were included in the table. ^b^—Out of 54,647 dyslipidemia patients who had LDL gap, 76% had Surescripts dispense data and were included in the table. ^c^—Out of 31,279 T2DM patients who had HBA1c gap, 74% had Surescripts dispense data and were included in the table. ^d^—Includes 3834 individuals who did not have a Sutter physician order for a medication but did have a Surescripts adjudication record for a medication in the specific class. ^e^—Includes 799 individuals who did not have a Sutter physician order for a medication but did have a Surescripts adjudication record for a medication in the specific class.

## Data Availability

Data are available on request due to restrictions, e.g., privacy or ethical reasons. The data (in summary statistics, non-identifiable aggregated form) presented in this study are available on request from the corresponding author. The data are not publicly available due to patient privacy considerations that prohibit the sharing of patient-level data.

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
