# Peer review of "Persistent Cardiometabolic Health Gaps: Can Therapeutic Care Gaps Be Precisely Identified from Electronic Health Records"

_healthcare, 2021, doi:10.3390/healthcare10010070_

Round 1

Reviewer 1 Report

  • This paper presented here highlights the value of using structured EHR to identify the health and care gaps of patients in primary care settings.  This approach can aid in the implementation of new algorithms or a clinical decision support tool to close these care gaps.
  • The concept of using structured EHR for optimizing patient care and/or quality improvement projects is not novel. This area has been extensively studied and used every day in clinical practice. Nonetheless, the role of natural language processing or AI in extracting the data/processing is something that is needed. 

  •  

    Regarding the results presented, it was not clear how the analysis accounted for the variability seen in managing patients with CM differently because of their race/ethnicity? 

  •  

    Assessing and characterizing medication adherence can be a challenging task in the absence of progress notes. Commonly, patients can take a long time to achieve a stable dose of medication. In addition, dose tapering could another challenge for patients who can't tolerate certain doses due to genetic causes or possible drug interactions.

  • Other suggestions would be to expand the introduction and give more rationale for the abstract. The abstract appears to focus only on the objective and results. 

Author Response

Point 1: This paper presented here highlights the value of using structured EHR to identify the health and care gaps of patients in primary care settings.  This approach can aid in the implementation of new algorithms or a clinical decision support tool to close these care gaps.

Response 1: Thank you for your insights on future applications of this work.

Point 2: The concept of using structured EHR for optimizing patient care and/or quality improvement projects is not novel. This area has been extensively studied and used every day in clinical practice. Nonetheless, the role of natural language processing or AI in extracting the data/processing is something that is needed.  

Response 2: We agree on both points. But what is lacking in the literature is a deeper understanding of how to operationalize use of structured data to readily and easily identify health gaps and care gaps, to integrate an understanding of medication adherence as a mediator of health gaps, and to reveal areas of clinical inertia. Poor medication adherence is a dominant mediator of patient health outcomes. While structured data on medication adherence is now widely available and stored in the EHR, many clinicians are still not fully aware of the presences of these data, and most do not know how to use it during encounters. Our paper reveals how the data can be used analytically to identify patients who have health gaps that are explained by low medication adherence. Our intention is to reveal how such analytics can be used to directly manage medication adherence through population health protocols or clinician messaging directly to patients. We agree that applying natural language processing to progress note text data could offer more insights on non-adherence and facilitate understanding of how to close health and care gaps. But this topic is beyond the focus of our paper. We also recognize that most healthcare systems in the US have limited analytic and IT resources to manage population health and that these health systems will strongly limit what they do to work that are technically and analytically straightforward. The motivation for our analysis in this paper derives from our own experience in working with clinical practices and more directly addressing their needs. We have elaborated on these points in the discussion section.

Point 3: Regarding the results presented, it was not clear how the analysis accounted for the variability seen in managing patients with CM differently because of their race/ethnicity?  

Response 3: We adjusted for race/ethnicity in the logistic regression models that were used to assess the association between medication adherence and each health gap to estimate the average effect cross all race/ethnicity groups. We did not specifically describe race/ethnicity differences in Table 4 but agree that it is worth noting. We have added 2 sets of new supplemental tables, Supplemental Table B1 showed the association (i.e., OR and 95%CI) between health gap and medication adherence for each race group. Supplemental tables B2a-B2c explicitly compare demographics, disease burdens, and other baseline factors by care gap status (had care gap, no care gap, and uncertain). We observed differences in demographics and other baseline factors across care gap status, and added to the result and discussion sections.

Point 4: Assessing and characterizing medication adherence can be a challenging task in the absence of progress notes. Commonly, patients can take a long time to achieve a stable dose of medication. In addition, dose tapering could another challenge for patients who can't tolerate certain doses due to genetic causes or possible drug interactions.

Response 4: We agree that there are important nuances to understanding why a patient is not adhering. Evidence indicates that the dominant reasons are, in order, 69% due to inattention (forgetfulness) and inertia, 16% due to cost, and 15% due to clinical concerns (e.g. side effects, medication not working) (https://www.pillsy.com/articles/medication-adherence-stats). Nonetheless, we agree that it is important to note other challenges that patients face in treatment adherence as you have noted and we have expanded in these points in the discussion section.  Thank you for the insights.

Point 5: Other suggestions would be to expand the introduction and give more rationale for the abstract. The abstract appears to focus only on the objective and results. 

Response 5: We agree and have expanded on this point in the abstract and the discussion section to emphasize the motivation of the study.

Reviewer 2 Report

This study was a clinical retrospective study, which aimed to determine the extent to which structured EHR data can be used to readily identify health and therapeutic care gaps for CM patients. This study selected medication adherence as a disease modifier to help understand if structured EHR data can provide insight into whether the underlying care gap is attributable to the physician or to the patient, as this attribution provides a basis for acting to close the gap. The authors concluded that structured EHR data can identify health gaps and a majority of care gaps, and that actions to close health gaps vary significantly at different medication adherence levels. This reviewer agrees with the authors that using structured EHR data to identify health gaps and measure medication adherence can be a first step in identifying factors attributable to care gaps and thereby help guide direction for clinical action. Uncertainty about the clinical actions best suited to close health gaps can be minimized by three essential components: efficient presentation of EHR structured 356 data; integration of patient-reported data or progress notes to indicate patient preferences, and translation of clinical guidelines to inform personalized recommendations. This reviewer considers that the concept of this study was interesting and clinically important, but this reviewer has some comments in this paper.

Major comments:

  1. Figure 2. Figure legend showed Figure 2, but the figure itself said “Figure 1”. Should be corrected.
  2. This is the critical issue of this study. The authors collected the clinical data of blood pressure, lipid and glucose (including HbA1c) levels. It was unclear how the authors examined the clinical data. They should clearly show in the Methods section.
  3. Further, the authors should show the clinical data in the Results section and Table(s).

Author Response

Point 1: Figure 2. Figure legend showed Figure 2, but the figure itself said “Figure 1”. Should be corrected.

Response 1: We agree and have made the correction accordingly to the Figure legend.

Point 2: This is the critical issue of this study. The authors collected the clinical data of blood pressure, lipid and glucose (including HbA1c) levels. It was unclear how the authors examined the clinical data. They should clearly show in the Methods section.

Response 2: Thanks for the comment. We have made following changes: 1) a supplemental material (supplemental Material Section A) is added to detail the criteria we used to identify each of the CM conditions. 2) we revised Material and Method section to describe how the EHR is used to identify eligible patients (Study Design and Study Population), identification of health gap and care gaps (Study outcomes), and derive medication adherence measures (Study Outcomes); 3) we also specify which subcohort is used for each analysis (Statistical Analysis). 

Point 3: Further, the authors should show the clinical data in the Results section and Table(s).

Response 3: Thanks for the comment. We have made following changes: 1) added details in describing prevalence of health gaps, care gaps, and medication adherence in the Results section as you suggested; 2) we added supplemental tables (Supplemental Table B2a-B2c) to compare the baseline characteristics across different care gap status (had care gap, no care gap, and uncertain); 3) we added a sentence to summarize the findings in above supplemental tables in the Results section.

Reviewer 3 Report

  1. Pop’n is not standard abbreviation to use.
  2. In the Flowchart "some time" between... is not correct in grammar
  3. Why the diagnosis of diabetes type II in this study used "Hb A1C >/= 7.0"? This is not correct for the diagnosis of diabetes. 
  4. What is statistical software that is used in this study?
  5. Table 3 and Table 4 should be moved to result section, not statistical analysis.
  6. What is the implication of this study? and What is future direction?

Author Response

Point 1: Pop’n is not standard abbreviation to use.

Response 1: We agree and have made corrections accordingly.  

Point 2: In the Flowchart "some time" between... is not correct in grammar

Response 2: We agree and have changed the text to read “35+ years of age at the first primary care visit between 10/01/2013-9/30/2015.”

Point 3: Why the diagnosis of diabetes type II in this study used "Hb A1C >/= 7.0"? This is not correct for the diagnosis of diabetes. 

Response 3: The reason we use HbA1c >=7.0 as cutoff is because we used ADA guideline 2019 which indicated that “A reasonable A1C goal for many nonpregnant adults is <7% (53 mmol/mol)”. We added a link to the guideline (https://care.diabetesjournals.org/content/42/Supplement_1/S61) to table 2.

Point 4: What is statistical software that is used in this study?

Response 4: We used SAS Enterprise Guide 7.1 (Cary, NC) for all analyses and have updated the methods section with this information.

Point 5: Table 3 and Table 4 should be moved to result section, not statistical analysis.

Response 5: We agree and have made the changes accordingly.

Point 6: What is the implication of this study? and What is future direction?

Response 6: The study has used three cardio-metabolic conditions to illustrate the limitation of using structured EHR to reveal health gaps, and therapeutic care gaps, we specifically focused on showing how to use medication adherence, an effect modifier to therapeutic care gaps, to attribute the care gap to clinicians or patients. Given medication adherence has not been commonly used in clinical decision making, the study provides insight on what information is needed and how to use the readily available medication adherence data in patient management. In a future study, we will test effectiveness of displaying medication adherence (using clinical decision tool) to clinicians in closing care gaps and improving disease control.

Round 2

Reviewer 1 Report

Thanks for the revisions and the well-articulated responses to my earlier comments. 

Reviewer 2 Report

This reviewer has no further comment.